# A Case of Biliary Cast Syndrome with Cholangiocarcinoma-like Lesion in a Patient with No History of Liver Transplantation

**DOI:** 10.3390/medicina59071272

**Published:** 2023-07-09

**Authors:** Sa-Hong Jo, Ho-Cheol Choi, Sung-Eun Park, Jin-Il Moon, Jung-Ho Won, Jae-Boem Na, Yang-Won Kim, Won-Jeong Yang, Byeong-Ju Koo, Jae-Kyeong Ahn, Seong-Je Kim

**Affiliations:** 1Department of Radiology, Gyeongsang National University School of Medicine and Gyeongsang National University Changwon Hospital, Changwon 51472, Republic of Korea; jo452y@naver.com (S.-H.J.);; 2Department of Radiology, Gyeongsang National University School of Medicine and Gyeongsang National University Hospital, Jinju 52727, Republic of Korea; 3Department of Internal Medicine, Gyeongsang National University School of Medicine and Gyeongsang National University Hospital, Jinju 52727, Republic of Korea

**Keywords:** biliary cast syndrome, biliary cast syndrome without liver transplantation, biliary cast syndrome with cholangiocarcinoma-like lesion

## Abstract

*Background and Objectives*: Biliary cast syndrome, which was first reported in 1975, is a rare disease that occurs after liver transplantation. The incidence is even lower in patients who have not undergone liver transplantation. This study reports a rare case of biliary cast syndrome with cholangiocarcinoma-like lesions in a patient who did not undergo liver transplantation. *Case Report:* Herein, we report a case of a 69-year-old man with right upper quadrant pain and elevated levels of alkaline phosphatase and gamma-glutamyl transferase, who had a history of total gastrectomy for gastric cancer and laparoscopic cholecystectomy for acute cholecystitis. Computed tomography (CT) revealed longitudinal bile stones in the extrahepatic and intrahepatic bile ducts and abrupt narrowing of the left main bile duct accompanied by a narrowing of the upstream bile duct in the left lobe of the liver. Based on the CT findings, the removal of the bile stones in the bile duct and additional examinations of the suspected cholangiocarcinoma were performed. The patient’s symptoms improved, and examinations for suspected cholangiocarcinoma showed no abnormal findings, and he was discharged one month later. *Conclusions*: The purpose of this case report is to share a rare case of Biliary Cast Syndrome (BCS) occurring without liver transplantation. Additionally, the report aims to share image findings that mimic cancer in BCS, with the goal of reducing unnecessary repetitive biopsies, minimizing patient discomfort, and decreasing unnecessary costs by aiding in the diagnosis of BCS.

## 1. Introduction

A cast in the bile duct is termed a biliary cast [1], and the formation of biliary casts was first described in 1975 [2]. This complication is most commonly reported in patients after liver transplantation [3]. Although the mechanism of biliary cast formation has not been elucidated, it is believed that biliary casts can be caused by damage to the mucous membrane of the bile duct, ischemia, bile duct infection, or foreign body reactions due to an indwelling T-pipe or stent after liver transplantation. Biliary casts are rarely observed in patients who have not undergone liver transplantation [1,3,4]. However, there have been reports of biliary cast formation in patients who have undergone liver or biliary tract surgery, long-term fasting or total parenteral nutrition, or large blood transfusions and in patients with severe infection during the treatment of disease [1,2,3,4,5,6,7].

In this case report, a patient that did not undergo liver transplantation and was diagnosed with biliary cast syndrome and cholangiocarcinoma-like lesions is presented.

## 2. Case Report

A 69-year-old man presented to the hospital with complaints of right upper quadrant pain. The patient had a history of total gastrectomy for gastric cancer eight years prior to presentation and laparoscopic cholecystectomy for acute cholecystitis one year prior to presentation. The patient had no other underlying diseases. On exam, a clear consciousness and an acute illness were noted. The patient’s blood pressure was 124/86 mmHg, heart rate was 92 beats/min, respiration rate was 20 breaths/min, and temperature was 36.5 °C. An abdominal examination revealed persistent right upper quadrant pain that worsened upon coughing. Other accompanying symptoms included nausea, dyspepsia, and chills.

At the time of admission, the patient’s leukocyte count was 6940/mm^3^, hemoglobin level was 10.1 g/dL, and platelet count was 364,000/mm^3^. His total protein level was 7.8 g/dL, albumin level was 3.6 g/dL, total bilirubin level was 0.89 mg/dL, direct bilirubin level was 0.65 mg/dL, aspartate aminotransferase level was 27 IU/L, alanine transaminase level was 21 IU/L, alkaline phosphatase level was 655 IU/L, gamma-glutamyl transferase level was 564 IU/L, serum urea nitrogen level was 17.7 mg/dL, creatinine level was 1.38 mg/dL, sodium level was 135.9 mEq/L, potassium level was 4.9 mEq/L, and C-reactive protein level was 45.9 mg/dL.

Computed tomography (CT) revealed longitudinal bile stones in the extrahepatic and intrahepatic bile ducts and abrupt narrowing of the left main bile duct accompanied by a narrowing of the upstream bile duct in the left lobe of the liver (Figure 1). Based on the CT findings, the removal of the bile stones in the bile duct and additional examinations of the suspected cholangiocarcinoma were planned.

To remove the bile stones, a right percutaneous transhepatic biliary drain was placed (Figure 2). The removed bile stones were longitudinal, brown, and hardened (Figure 3) [2]. Linear bile stones that could not be removed were observed on magnetic resonance imaging (MRI) that was obtained to confirm the suspected cholangiocarcinoma site. These bile stones had hypointense signals on T2-weighted images and hyperintense signals on T1-weighted images (Figure 4) [8,9].

The suspected cholangiocarcinoma site was examined via MRI. The MRI revealed abrupt narrowing of the left main bile duct accompanied by a narrowing of the upstream bile duct in the left lobe of the liver, which was consistent with the CT findings. These findings suggest cholangiocarcinoma with hyperintense signals on T2-weighted images in the surrounding segment (segment 4) of the right hepatic lobe (Figure 4). A biopsy of the left intrahepatic biliary narrowing site, which was the suspected site of cholangiocarcinoma identified on CT and MRI, was conducted via endoscopic retrograde cholangiopancreatography and left percutaneous transhepatic biliary drainage. Two additional biopsies of segment 4 of the right hepatic lobe were obtained under ultrasound guidance. The biopsies revealed findings of chronic inflammation. There were no pathological findings suggestive of cholangiocarcinoma.

## 3. Discussion

Biliary cast syndrome has been reported to occur after liver transplantation. However, biliary cast syndrome was diagnosed in this patient with a history of gastric cancer surgery and cholecystectomy who did not undergo liver transplantation. Various types of bile duct damage can occur after cholecystectomy [10] and are thought to cause biliary cast syndrome [1,2,3,4,5]. Therefore, a history of cholecystectomy, not liver transplantation, may play a role in the development of biliary cast syndrome.

Bile duct stenosis has been reported after liver transplantation [9]. The presence of cholangiocarcinoma-like lesions in a patient with biliary cast syndrome who did not undergo liver transplantation is unusual. Bile duct damage after cholecystectomy may be observed at the confluence of the left and right main bile ducts [10]. This damage looks similar to cholangiocarcinoma lesions, as in the patient in this case report. These findings suggest that cholangiocarcinoma-mimicking lesions may be observed in patients with biliary cast syndrome without a history of liver transplantation.

The symptoms of biliary cast syndrome are mainly caused by cholangitis due to the blockage of the bile ducts. Rarely, bile duct obstruction test results are elevated without clear symptoms of cholangitis [11]. The symptoms of cholangitis were unclear in the patient in this case report. There was no direct increase in the patient’s bilirubin level, though the patient had right upper quadrant pain and elevated alkaline phosphatase and gamma-glutamyl transferase levels, which are observed in patients with bile duct obstruction. These rare symptoms and laboratory findings indicate that the test results may suggest bile duct obstruction prior to the manifestation of cholangitis symptoms in patients with biliary cast syndrome.

Several tests were conducted to check for cholangiocarcinoma-like lesions in this patient. Furthermore, the biliary casts had a typical appearance on several imaging examinations. After removal, the biliary casts were brown with a linear or branched pattern [2]. On CT images, the biliary casts are presented in the shape of hyperdense linear materials. On MRI, the linear biliary casts had hypointense and hyperintense signals on T2-weighted and T1-weighted images, respectively [8,9]. On fluoroscopy, the casts appeared as linear, filling defects inside the bile duct.

The treatment of Biliary Cast Syndrome (BCS) typically lacks a standardized approach and depends on factors such as bile obstruction and liver function. It is commonly managed through a combination of medical and interventional methods. Endoscopic or percutaneous interventions are frequently employed to remove biliary casts, improve bile flow surgically, or perform hepatic lobe resection or liver transplantation when necessary [1,3,4,6,7,9,11,12,13]. In the case of this patient with a history of total gastrectomy, percutaneous intervention was undertaken to establish percutaneous biliary drainage, followed by multiple procedures for the removal of stones.

## 4. Conclusions

The purpose of this case report is to share a rare case of Biliary Cast Syndrome (BCS) occurring without liver transplantation. Additionally, the report aims to share image findings that mimic cancer in BCS, with the goal of reducing unnecessary repetitive biopsies, minimizing patient discomfort, and decreasing unnecessary costs by aiding in the diagnosis of BCS.

## Figures and Tables

**Figure 1 medicina-59-01272-f001:**
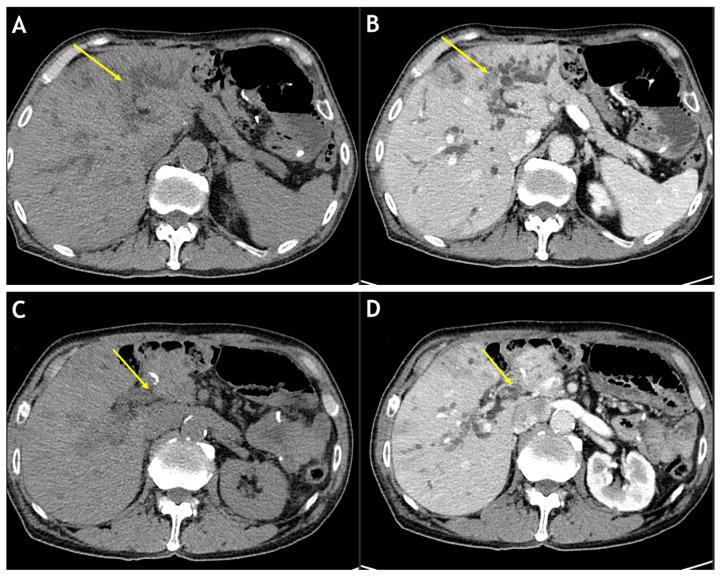
Computed tomography. The pre-contrast, axial computed tomography (CT) image (**A**) and contrast-enhanced, axial CT image (**B**) reveal abrupt narrowing (arrows) in the left main intrahepatic duct with disproportionate upstream biliary duct dilatation. The pre-contrast, axial CT image (**C**) and contrast-enhanced, axial CT image (**D**) reveal hyperdense materials (arrows) in the common bile duct. A contrast-enhanced, coronal CT image (**E**) reveals intraductal hyperdense materials (arrows) in the common bile duct.

**Figure 2 medicina-59-01272-f002:**
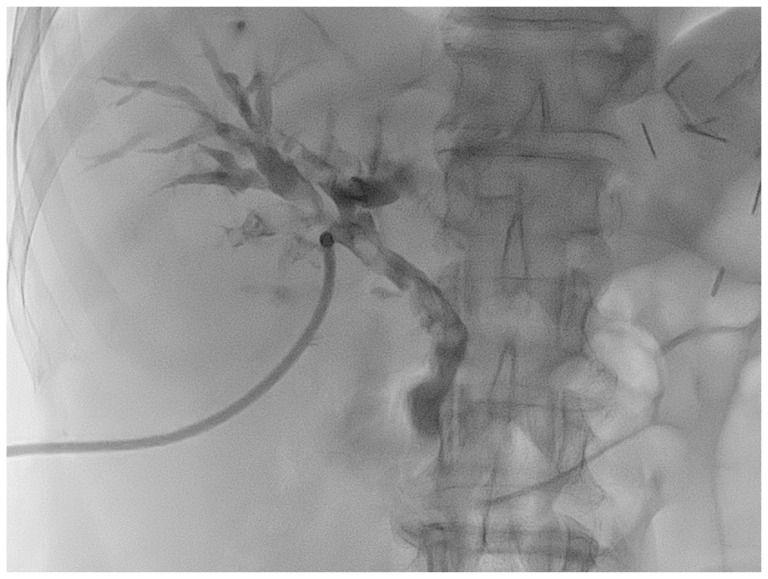
Cholangiography reveals multiple filling defects in the common bile duct and intrahepatic ducts with diffuse extrahepatic and intrahepatic duct dilatation. Disproportionate dilatation is noted in the left intrahepatic duct.

**Figure 3 medicina-59-01272-f003:**
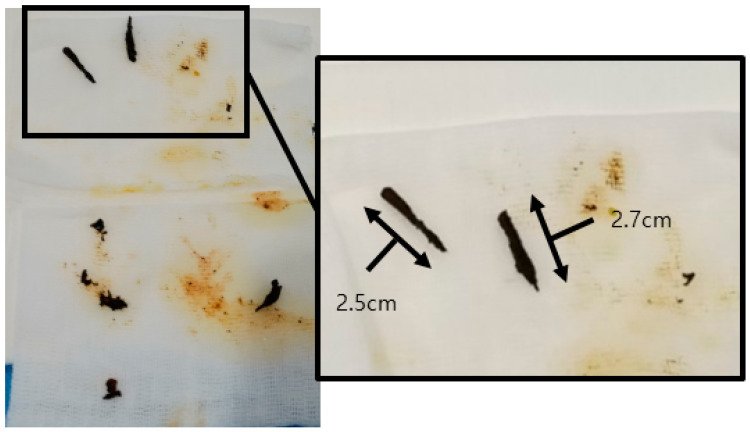
Percutaneous transhepatic biliary drainage was conducted to remove the bile stones. The stones are linear, branching, brown, and hardened.

**Figure 4 medicina-59-01272-f004:**
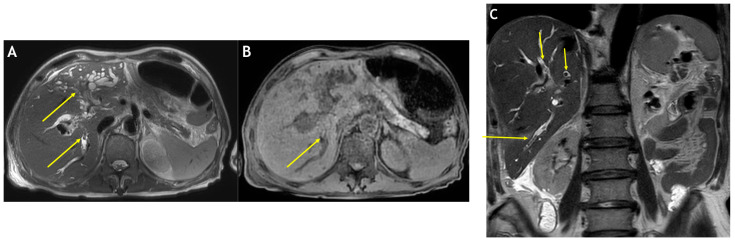
Several linear intrahepatic materials (arrows) have a hypointense signal on axial T2-weighted magnetic resonance imaging (**A**) and a hyperintense signal on axial T1-weighted magnetic resonance imaging (**B**). A coronal, T2-weighted image (**C**) reveals linear intraductal materials (arrows) with hyperintense signals.

## Data Availability

Not applicable.

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
