# Peer review of "A Case of Biliary Cast Syndrome with Cholangiocarcinoma-like Lesion in a Patient with No History of Liver Transplantation"

_medicina, 2023, doi:10.3390/medicina59071272_

Round 1

Reviewer 1 Report

I would like to learn the general reasons about the occurrence of biliary cast syndrome. Have you made the arterial visualization of the liver? Did anything happened unexpectedly during cholecystectomy? Maybe hepatic artery and /or its branches have accidentally damaged during cholecystectomy? Do you have any information about the past surgical procedures? I think the authors did not explain why did the biliary cast syndrome occurred in this patient. What messages do the authors tell the readers about this case report? what should we do in order to protect these patients from biliary cast syndrome?

it is fine. but it can be improved.

Author Response

 We would like to thank the Editor and the Reviewers for their meticulous review of our manuscript and for their insightful comments, which we believe have helped us provide a more balanced account of our research. We have carefully revised our manuscript according to your suggestions and made the necessary changes. Please find below our point-by-point responses to all comments.

Response to Reviewer 1 Comments

Point 1: I would like to learn the general reasons about the occurrence of biliary cast syndrome.

Response 1: According to References 1, 3, and 4 of this case report, it is believed that biliary casts can be caused by damage to the mucous membrane of the bile duct, ischemia, bile duct infection, or foreign body reactions resulting from the presence of an indwelling T-pipe or stent after liver transplantation, although the mechanism of biliary cast formation has not been elucidated.

(I did not provide additional information regarding the common causes of BCS as they are already mentioned in the introduction)

Point 2: Have you made the arterial visualization of the liver?

Response 2: I'm sorry, but angiography was not performed. However, based on the CT and MRI scans of the patient's case, there doesn't appear to be any abnormal findings in the arteries.

Point 3: Did anything happened unexpectedly during cholecystectomy? Maybe hepatic artery and /or its branches have accidentally damaged during cholecystectomy?

Response 3: We would like to thank you for your insightful comment. According to the surgical records of the patient's laparoscopic cholecystectomy, there was no mention of any vascular damage related to the surgery. Additionally, the follow-up CT scans conducted after the surgery did not show any distinct imaging findings of arterial injury.

Point 4: Do you have any information about the past surgical procedures? I think the authors did not explain why did the biliary cast syndrome occurred in this patient.

Response 4: We would like to thank you for your valuable comment. The patient in the case report had undergone a gastric resection surgery 8 years ago for gastric cancer and a laparoscopic cholecystectomy surgery 1 year ago for acute cholecystitis. This information was obtained through the patient's medical questionnaire and referral letters from other hospitals. However, there is no additional information available regarding the specific details of the surgical procedures.

According to the references, although the exact cause of biliary cast syndrome (BCS) in liver transplant patients has not been clearly identified, certain factors such as damage to the mucous membrane of the bile duct, ischemia, bile duct infection, or foreign body reactions due to the presence of an indwelling T-pipe or stent after liver transplantation are considered potential causes. In cases where liver transplantation is not involved, reported cases do not clearly identify the exact cause, but they mention various potential factors in each case. These factors include head trauma (ICU care, gallbladder sludge), systemic autoimmune disorders (hepatic infarction and ischemia in antiphospholipid antibody syndrome), malignancy (fasting, TPN, biliary infection, ischemia in non-Hodgkin lymphoma and chronic lymphocytic leukemia), traveler's diarrhea and biliary infection, hematopoietic stem cell transplantation (resulting in bile duct stricture and immune reaction in myelodysplastic syndromes), brain surgery (longstanding bile sludge due to a bed-ridden state), and cholecystectomy (increased bile pigment load, fasting, TPN, biliary infection). Considering these factors, possible causes in this case could include the increased bile pigment load resulting from cholecystectomy and the possibility of damage to the mucous membrane of the bile duct and/or ischemia (which may not be visible on imaging).

(I have updated references to provide support for the information mentioned above regarding cases of Biliary Cast Syndrome (BCS) occurring without liver transplantation. Reference 10-11)

Point 5: what should we do in order to protect these patients from biliary cast syndrome?

Response 5: We would like to thank you for your insightful comment. While it would be ideal to prevent the occurrence of biliary cast syndrome, the primary objective of this case report is to share a rare case of biliary cast syndrome occurring without liver transplantation and to provide image findings that mimic cancer in BCS. The aim is to reduce unnecessary biopsies, minimize patient discomfort, and decrease unnecessary costs by aiding in the diagnosis of biliary cast syndrome.

Additionally, in order to protect patients from bile duct cast syndrome, it is important to consider potential causes such as bile pigment load resulting from cholecystectomy and the possibility of damage to the mucous membrane of the bile duct and/or ischemia (which may not be visible on imaging). Efforts should be made to minimize bile duct injury and vascular damage during cholecystectomy, and post-cholecystectomy measures may be necessary to reduce increased bile pigment loading.

(Based on the comment, I have revised and improved the conclusions in the abstract and the main body of the text. Line 29-33, 161-165)

Reviewer 2 Report

Please provide better grammar connectors in your introduction. Adequate case development, but you need some minors grammar corrections. 

Please provide better grammar connectors in your introduction. Adequate case development, but you need some minors grammar corrections. 

Author Response

 We would like to thank the Editor and the Reviewers for their meticulous review of our manuscript and for their insightful comments, which we believe have helped us provide a more balanced account of our research. We have carefully revised our manuscript according to your suggestions and made the necessary changes. Please find below our point-by-point responses to all comments.

Response to Reviewer 2 Comments

Point 1: Please provide better grammar connectors in your introduction. Adequate case development, but you need some minors grammar corrections.

Response 1: We would like to thank you for your valuable comment. Based on the comment, I have made changes to the introduction to address awkward expressions.

 (Line 38 : biliary cast formation --> the formation of biliary casts

Line 41 : biliary casts may be --> it is believed that biliary casts can be

Line 44-45 : , though biliary cast formation has been reported in patients who underwent --> . However, there have been reports of biliary cast formation in patients who have undergone)

Reviewer 3 Report

The authors present an unusual case of biliary cast syndrome in a patient that has not undergone hepatic transplant. The case is well structured and clearly presented. The imagistic is suggestive.

However the Discussion could be enlarged, with a paragraph regarding the management of such cases, and the references updated accordingly

Author Response

 We would like to thank the Editor and the Reviewers for their meticulous review of our manuscript and for their insightful comments, which we believe have helped us provide a more balanced account of our research. We have carefully revised our manuscript according to your suggestions and made the necessary changes. Please find below our point-by-point responses to all comments.

Response to Reviewer 3 Comments

Point 1: The authors present an unusual case of biliary cast syndrome in a patient that has not undergone hepatic transplant. The case is well structured and clearly presented. The imagistic is suggestive. However the Discussion could be enlarged, with a paragraph regarding the management of such cases, and the references updated accordingly.

Response 1: We would like to thank you for your insightful comment. Based on your comment, I now understand that including a discussion of the patient's treatment in the case report would be more beneficial. I have added a paragraph to the discussion section regarding the general treatment of Biliary Cast Syndrome (BCS) and the treatment approach used in this particular case. I have also updated the references accordingly.

(Line 152-159 : added a paragraph to the discussion section

Reference 11-13 : updated the references)

Round 2

Reviewer 1 Report

congratulations for your effort!

it is OK!

Reviewer 3 Report

The authors revised the manuscript according to the comments.